# Uni3D: Exploring Unified 3D Representation at Scale

Junsheng Zhou[1,2*]    Jinsheng Wang[1*]    Baorui Ma[1*]    Yu-Shen Liu[2]

Tiejun Huang[1,3]    Xinlong Wang[1]

[1] Beijing Academy of Artificial Intelligence    [2] Tsinghua University    [3] Peking University

Code & Models: https://github.com/baaivision/Uni3D

## Abstract

Scaling up representations for images or text has been extensively investigated in the past few years and has led to revolutions in learning vision and language. However, scalable representation for 3D objects and scenes is relatively unexplored. In this work, we present Uni3D, a 3D foundation model to explore the unified 3D representation at scale. Uni3D uses a 2D initialized ViT end-to-end pretrained to align the 3D point cloud features with the image-text aligned features. Via the simple architecture and pretext task, Uni3D can leverage abundant 2D pretrained models as initialization and image-text aligned models as the target, unlocking the great potential of 2D models and scaling-up strategies to the 3D world. We efficiently scale up Uni3D to one billion parameters, and set new records on a broad range of 3D tasks, such as zero-shot classification, few-shot classification, open-world understanding and part segmentation. We show that the strong Uni3D representation also enables applications such as 3D painting and retrieval in the wild. We believe that Uni3D provides a new direction for exploring both scaling up and efficiency of the representation in 3D domain.

## 1 Introduction

3D representation learning is one of the most fundamental problems in 3D computer vision, especially with the rapid development of 3D sensors (e.g., LiDAR) and the growing demands in real-world applications, e.g., autonomous driving, augmented/virtual reality and robotics. Existing methods make great progress in 3D model architecture (Qi et al., 2017a;b; Yu et al., 2021; Wang et al., 2019), learning objective (Yu et al., 2022; Wang et al., 2021), task-oriented modeling (Zhou et al., 2020; Yin et al., 2021; Zhao et al., 2021; Zhou et al., 2022a; Ma et al., 2023a), etc. However, most of the works explore at a relatively small scale, with limited parameters, data, and task scenarios. Learning scalable 3D representation that can transfer in the wild is relatively unexplored and remains a challenging problem.

In the past few years, scaling up pre-trained language models (Brown et al., 2020; Liu et al., 2019; Raffel et al., 2020) has largely revolutionized natural language processing. Some recent works (Radford et al., 2021; Dosovitskiy et al., 2020; Bao et al., 2021; He et al., 2022; Fang et al., 2022) translate the progress from language to 2D vision via model and data scaling. Motivated by their success, it is appealing that we can also lift this success from 2D to 3D, i.e., to learn a scalable 3D representation model that can transfer in the 3D world. Recently, as the release of a large-scale 3D dataset Objaverse (Deitke et al., 2023b), a few works have tried to explore scalable pretraining in 3D, but either still limit to the small-scale 3D backbones (Xue et al., 2023a;b), or can hardly scale to a relatively larger size (Liu et al., 2023), e.g., 72M in Fig. 1.

In this work, we propose Uni3D, a unified and scalable 3D pretraining framework for large-scale 3D representation learning, and explore its limits at the scale of one billion parameters with a million 3D shapes and 10 million images paired with 70 million texts. Uni3D uses a 2D ViT as the 3D encoder initialized with the best 2D prior, which is then end-to-end pre-trained to align the 3D point

---

*Equal contribution. Correspondence to {*brma@baai.ac.cn*} and {*wangxinlong@baai.ac.cn*}.

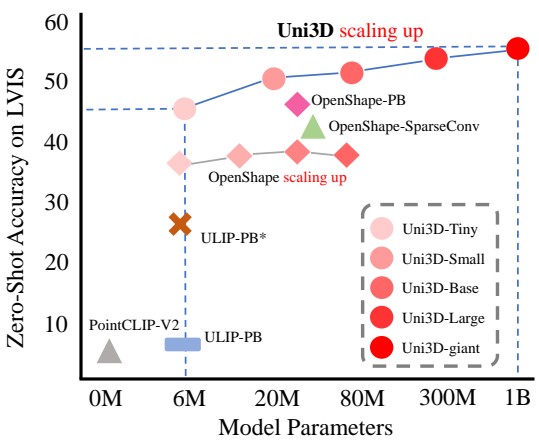

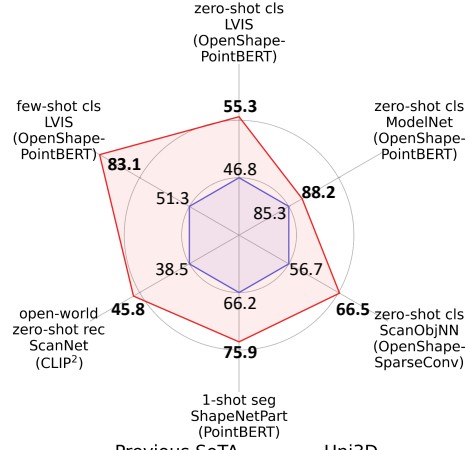

Figure 1: The parameter and zero-shot accuracy comparison. Uni3D scales up 3D representation from 6M to 1B. 'PB' indicates PointBERT. 'ULIP-PB*' indicates retrained ULIP on the ensambled large dataset.

Figure 2: Qualitative comparisons between Uni3D and previous SoTA methods under different tasks and benchmarks. The scale of each axis in the radar chart is normalized by the performance of previous SoTAs .

cloud features with the image-text aligned features. Via the simple architecture and pretext task, Uni3D can leverage abundant 2D pre-trained models as initialization (Fang et al., 2022; Caron et al., 2021), and image-text aligned models as the target (Radford et al., 2021; Sun et al., 2023; Cherti et al., 2023), unlocking the great potential of 2D models and scaling-up strategies to the 3D world.

In addition, we systematically study the scalability and flexibility of Uni3D in terms of 1) model scaling from 6M to 1B parameters, 2) 2D initialization from visual self-supervised to text supervised, and 3) text-image aligned target model from 150M to 5B parameters. We observe continuous performance improvements as the scaling of each component under the flexible and unified framework. The sharable 2D prior and scale-up strategies also largely benefit the large-scale 3D representation learning.

For the first time, we demonstrate a billion-scale 3D representation model that transfers well to various downstream tasks and scenarios. As shown in Fig. 2, Uni3D yields a boost compared to prior art in various zero-shot and few-shot 3D tasks. Specifically, Uni3D achieves a zero-shot classification accuracy of 88.2% on ModelNet, which surprisingly performs on par with some supervision methods. Uni3D also achieves state-of-the-art performance on other representative 3D tasks such as open-world understanding, part segmentation, etc. In addition, we present some interesting applications with the strong 3D representation learned by Uni3D, such as point cloud painting and text/image-based 3D shape retrieval.

By scaling up 3D foundation models with a simple and unified pre-training to learn strong 3D representation across tasks, we hope Uni3D would bridge the gap between 2D and 3D vision, and contribute to the big convergence across different modalities. To facilitate future research, we will release all the code and 3D foundation models.

## 2 RELATED WORK

**3D Representation Learning.** Learning representations from point clouds for 3D understanding (Qi et al., 2017a;b; Wang et al., 2019; Zhou et al., 2024a) has been fully explored in recent years, and have shown great potential in 3D applications (Zhou et al., 2023a; Wen et al., 2022; Ma et al., 2021a; Li et al., 2023a; Ma et al., 2023c; Zhou et al., 2023b; 2022b; 2024b; Li et al., 2024; Ma et al., 2023b; 2022b;a; Jin et al., 2023). Some works further studied self-supervised pretraining for point clouds by specific 3D pretext tasks like self-reconstruction (Wang et al., 2021) and mask point modeling (Yu et al., 2022). These works merely explore under limited 3D data (e.g. ShapeNet (Chang et al., 2015)) and do not investigate multi-modal representation from 2D/NLP to 3D.

With the recent success in learning visual concepts from raw text with contrastive learning like CLIP (Radford et al., 2021; Jia et al., 2021; Li et al., 2022; Ramesh et al., 2022; Gregoromichelaki et al., 2022), recent works (Liu et al., 2023; Qi et al., 2023; Xue et al., 2023a; Hegde et al., 2023; Lei et al., 2023) seek to learn 3D representations by aligning text, image, and point cloud features through in a similar contrastive learning way. Recently, as the release of a large-scale 3D dataset Objaverse (Deitke et al., 2023b), OpenShape (Liu et al., 2023) and ULIP2 (Xue et al., 2023b) have tried to explore scalable pretraining in 3D, but either still limit to the small-scale 3D backbones (Xue et al., 2023b), or can hardly scale to a relatively larger size (Liu et al., 2023).

**Foundation models.** Recently, it has been drawing significant attention to design foundation models for unifying and scaling up representations under different modalities (e.g. NLP, 2D vision). Starting from NLP, recent works in scaling up pre-trained language models (Brown et al., 2020; Liu et al., 2019; Raffel et al., 2020) have largely revolutionized natural language processing. Some research in 2D vision (Radford et al., 2021; Dosovitskiy et al., 2020; He et al., 2022; Fang et al., 2022) translates the progress from language to 2D vision via model and data scaling. However, such a phenomenon has not been well-established and explored in the 3D domain, due to the limited 3D data and difficulties in unifying and scaling up 3D backbones. Meta-Transformer (Zhang et al., 2023) and FrozeCLIP (Huang et al., 2022b) have indicated a promising future for developing a unified framework with a modality-shared encoder. However, they require retraining task-specific heads with labor-intensive manual labeling of ground truth for different downstream tasks, which leads to a lack of out-of-domain capabilities. In this work, we design the first billion-scale 3D foundation model with a unified 3D representation. The unified ViT architecture allows us to simply scale up Uni3D with the well-studied 2D/NLP scaling-up strategies. We anticipate Uni3D to serve as a bridge between 2D and 3D vision, facilitating significant convergence across various modalities.

# 3 METHOD

We introduce Uni3D, a unified and scalable 3D pretraining framework for large-scale 3D representation learning by aligning 3D point cloud features with the image-text aligned features. The overview of Uni3D is shown in Fig. 3. We first present how we design, scale up and initialize a unified 3D representation in Uni3D in Sec. 3.1. We then introduce the multi-modal contrastive learning for aligning image and language with point cloud in Sec. 3.2. More training details are provided in Sec. A of the appendix.

## 3.1 UNIFIED 3D REPRESENTATION

Uni3D leverages a unified vanilla transformer structurally equivalent to 2D Vision Transformer (ViT) (Dosovitskiy et al., 2020) as the backbone. The only difference here is that we replace the patch embedding layer in ViT with a specific point tokenizer to achieve 3D embeddings. The point tokenizer keeps the same as PointBERT (Yu et al., 2022) to first group points into local patches with FPS (farthest point sampling) and kNN (k nearest neighbor), and then extract token embeddings with a tiny PointNet (Qi et al., 2017a) for each patch. The vanilla transformer is then applied to the 3D tokens to extract the 3D representations.

**Scaling Up Uni3D.** Previous works on point cloud representation learning merely focus on designing specific model architectures for pursuing better performances in different applications and are limited to a certain small-scale dataset (e.g. ShapeNet (Chang et al., 2015), ModelNet (Wu et al., 2015)). With the recent successes in large-scale 3D data (e.g. Objaverse (Deitke et al., 2023b;a)), a few recent works (Xue et al., 2023a; Liu et al., 2023; Xue et al., 2023b) have tried to explore scalable pretraining in 3D, but either still limit to the small-scale 3D backbones (Xue et al., 2023a), or can hardly scale to a relatively larger size (Liu et al., 2023). The difficulties lie in the un-unified backbones and pretraining in 3D domain, where each backbone requires a specific scaling-up strategy, which is rarely explored. Moreover, some backbones (e.g. PointMLP (Ma et al., 2021b), DGCNN (Wang et al., 2019)) require modeling local patterns completely on dense points, which brings extensive computational cost when scaling up.

We justify that Uni3D, which directly leverages the vanilla transformer structurally equivalent to ViT, can naturally solve the difficulties by simply scaling up the model size with the well-studied unified 2D/NLP scaling-up strategies. Specifically, we leverage the strategy of ViT which gradually

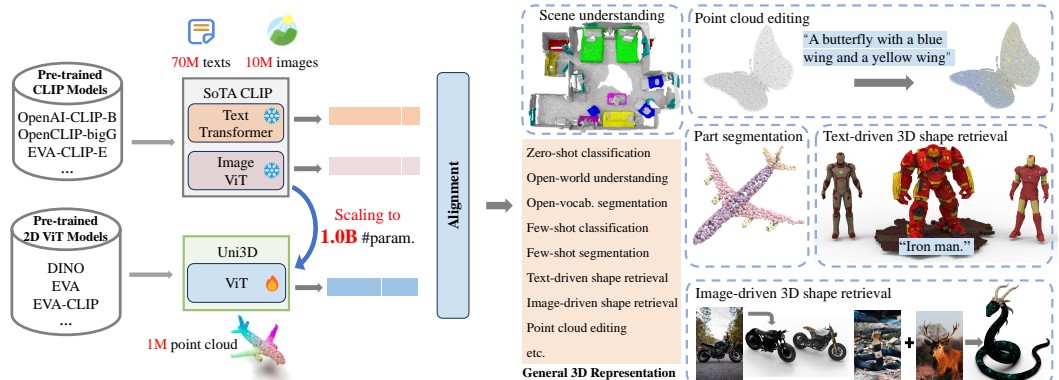

Figure 3: **The overview of Uni3D**. Uni3D is a unified and scalable 3D pretraining framework for large-scale 3D representation learning. We scale up Uni3D to one billion parameters with a million 3D shapes paired with 10 million images and 70 million texts. Uni3D uses a 2D ViT as the 3D encoder initialized with the best 2D prior from abundant 2D pre-trained models, which is then end-to-end pre-trained to align the 3D point cloud features with the image-text aligned ones from SoTA CLIP models. Uni3D shows superior performance on a wide range of benchmarks.

scales up Transformer from Tiny (6 M), Small (23M), Base (88 M), Large (307 M) to giant (1B) and replace the Transformer of Uni3D with different sizes of ViT as the scaled-up version of Uni3D at different model sizes. The effectiveness and efficiency of our scaling-up strategy are fully demonstrated by the comprehensive exploration of scaling up ViT in the 2D vision domain. As shown in Fig. 1 and Tab. 5, we observe continuous performance improvements as the scaling of model size under the flexible and unified framework.

Given the unified scaling-up strategy, we train the largest 3D presentation model with one billion parameters under the multi-modal alignment learning objective, in a large-scale dataset of nearly one million 3D shapes, along with paired 10 million images and 70 million texts. For the first time, we demonstrate a billion-scale 3D representation model that transfers well to various downstream tasks and scenarios.

**Initializing Uni3D.** Another challenge that prevents previous works in scaling up 3D backbones is that larger model sizes lead to overfitting and difficulties in convergence. A naive solution is to pretrain each 3D backbone with specific 3D pretext tasks (e.g. PointBERT (Yu et al., 2022), OcCo (Wang et al., 2021)), and leverage the pretrained parameters as the initialization. However, this results in expensive training costs, and the relatively limited scale of 3D data for pretraining makes it challenging to establish a robust initialization for stabilizing cross-modal contrastive learning.

In Uni3D, we directly leverage the vanilla transformer structurally equivalent to ViT as the 3D backbone, which brings a new perspective of introducing pretrained priors. Specifically, we can naturally adopt the pretrained large models in other modalities which share the same vanilla transformer as ours to initialize Uni3D, such as the 2D pretrained model DINO (Caron et al., 2021), EVA (Fang et al., 2022), EVA-02 (Fang et al., 2023) and the cross-modal models CLIP (Radford et al., 2021), EVA-CLIP (Sun et al., 2023), etc. These pretrained models are trained in datasets consisting of billions of images and texts, which already learn rich underlying representational abilities for Transformer and have the potential to enhance and stabilize the learning of large-scale 3D representations. Uni3D is not limited to a specific pretrained model for initialization, where we can flexibly leverage any off-the-shelf Transformer-based pretrained models at any modalities for pushing the performance and exploring the cross-modal pretraining (please refer to Sec. 4.7 for detailed analysis).

## 3.2 MULTI-MODAL ALIGNMENT

We train Uni3D to learn the multi-modal alignment across language, image and point cloud following a similar paradigm as ULIP (Xue et al., 2023a) and OpenShape (Liu et al., 2023).

**Datasets.** In order to keep the experimental settings consistent with other methods for a fair comparison, we adopt the ensembled 3D dataset provided by OpenShape for training, which consists of four 3D dataset, i.e., Objaverse (Deitke et al., 2023b), ShapeNet (Chang et al., 2015), 3D-FUTURE

Table 1: Zero-shot classification on Objaverse-LVIS (Deitke et al., 2023b), ModelNet40 (Wu et al., 2015), and ScanObjectNN (Uy et al., 2019). († represents the best results achieved on different benchmarks respectively)

| Method | training shape source | Objaverse-LVIS | | | ModelNet40 | | | ScanObjectNN | | |
|---|---|---|---|---|---|---|---|---|---|---|
| | | Top1 | Top3 | Top5 | Top1 | Top3 | Top5 | Top1 | Top3 | Top5 |
| ULIP-PointBERT | Ensembled (no LVIS) | 21.4 | 38.1 | 46.0 | 71.4 | 84.4 | 89.2 | 46.0 | 66.1 | 76.4 |
| OpenShape-SparseConv | | 37.0 | 58.4 | 66.9 | 82.6 | 95.0 | 97.5 | 54.9 | 76.8 | 87.0 |
| OpenShape-PointBERT | | 39.1 | 60.8 | 68.9 | 85.3 | 96.2 | 97.4 | 47.2 | 72.4 | 84.7 |
| **Uni3D** | | **47.2** | **68.8** | **76.1** | **86.8** | **97.3** | **98.4** | **66.5** | **83.5** | **90.1** |
| ULIP-PointBERT | Ensembled | 26.8 | 44.8 | 52.6 | 75.1 | 88.1 | 93.2 | 51.6 | 72.5 | 82.3 |
| OpenShape-SparseConv | | 43.4 | 64.8 | 72.4 | 83.4 | 95.6 | 97.8 | 56.7 | 78.9 | 88.6 |
| OpenShape-PointBERT | | 46.8 | 69.1 | 77.0 | 84.4 | 96.5 | 98.0 | 52.2 | 79.7 | 88.7 |
| **Uni3D** | | **53.5** | **75.5** | **82.0** | **87.3** | **98.1** | **99.2** | **63.9** | **84.9** | **91.7** |
| **Uni3D †** | | **55.3** | **76.7** | **82.9** | **88.2** | **98.4** | **99.3** | **65.3** | **85.5** | **92.7** |

(Fu et al., 2021) and ABO (Collins et al., 2022). We sample 10,000 points from the mesh surface with colors and render 10 color images from different views that uniformly cover the whole shape. The point cloud-text-image triplets are conducted in the same way as OpenShape.

**Objective.** The illustration of the multi-modal alignment is shown in Fig. 3. We initialize the Uni3D point encoder $f_P$ with pretrained 2D ViT models and obtain the text encoder $f_T$ and image encoder $f_I$ from CLIP models. We train $f_P$ to learn 3D representations by aligning them to well-learned 2D / Language representations of CLIP models and distills cross-modal knowledge. Both $f_I$ and $f_T$ are frozen since they are well-optimized, and only $f_P$ are learnable during training. Given a batch of $N$ triplets $\{(P_i, I_i, T_i)\}_{i=1}^{N}$, where $P_i$, $I_i$, $T_i$ donate a point cloud and its corresponding image and text obtained from the same 3D shape. We first achieve the normalized feature for the sampled triplets as $\{(e_i^P = f_P(P_i)/|f_P(P_i)|, e_i^I = f_I(I_i)/|f_I(P_i)|, e_i^T = f_T(T_i)/|f_T(T_i)|)\}_{i=1}^{N}$. The contrastive loss is then formulated as:

$$-\frac{1}{4N}\sum_{i=1}^{N}\left(\log\frac{\exp(e_i^P \cdot e_i^T/\tau)}{\sum_j \exp(e_i^P \cdot e_j^T/\tau)} + \log\frac{\exp(e_i^T \cdot e_i^P/\tau)}{\sum_j \exp(e_i^T \cdot e_j^P/\tau)} + \log\frac{\exp(e_i^P \cdot e_i^I/\tau)}{\sum_j \exp(e_i^P \cdot e_j^I/\tau)} + \log\frac{\exp(e_i^I \cdot e_i^P/\tau)}{\sum_j \exp(e_i^I \cdot e_j^P/\tau)}\right),$$
(1)

where $\tau$ is a learnable temperature. The training target is to minimize the triplet contrastive loss.

**Image-Text Aligned Target.** We further justify that Uni3D is not limited to a specific CLIP teacher, where we can switch it to off-the-shelf SoTA CLIP models with different model scales flexibly to achieve better performance. For example, we can simply change the CLIP source from OpenAI-CLIP (Radford et al., 2021), OpenCLIP (Cherti et al., 2023) to the best EVA-CLIP (Sun et al., 2023), and probably to the better CLIP in the future. We can also directly scale up the CLIP teacher from EVA-CLIP-B (150 M) to EVA-CLIP-E (5 B). This demonstrates the flexibility and scalability of Uni3D and shows the potential of Uni3D to progress with the progress of CLIP models.

## 4 EXPERIMENT

### 4.1 ZERO-SHOT SHAPE CLASSIFICATION

We first evaluate Uni3D under the zero-shot shape classification task. We conduct experiments under three benchmarks: ModelNet (Wu et al., 2015), ScanObjNN (Uy et al., 2019) and Objaverse-LVIS (Deitke et al., 2023b). ModelNet and ScanObjNN are widely-used datasets which contains 15 and 40 common categories, respectively. The Objaverse-LVIS benchmark is an annotated and cleaned subset of Objaverse which contains 46,832 shapes of 1,156 LVIS categories. We follow the settings of OpenShape (Liu et al., 2023) to conduct evaluations. For Objaverse-LVIS, we use 10,000 sampled colored points as input. For ModelNet40, we utilize 10,000 sampled points without color as input. For ScanObjNN, the input is 2,048 sampled points without color from the OBJ_ONLY version. We compare Uni3D with the previous SoTA methods in the zero-shot shape classification task, such as PointCLIP (Zhang et al., 2022), PointCLIP V2 (Zhu et al., 2022), ULIP (Xue et al., 2023a) and OpenShape (Liu et al., 2023). Note that PointCLIP and PointCLIP V2 directly project point clouds into images and leverage 2D CLIP for classification, while other methods adopt a similar schema to train a native 3D backbone for aligning 3D representations with image and text representations produced by a pretrained CLIP. We follow OpenShape (Liu et al., 2023) to report the performance

Table 2: Zero-shot recognition in ScanNet. Avg.: the average Top1 accuracy across all categories.

| Method | Avg. | Bed | Cab | Chair | Sofa | Tabl | Door | Wind | Bksf | Pic | Cntr | Desk | Curt | Fridg | Bath | Showr | Toil | Sink |
|---|---|---|---|---|---|---|---|---|---|---|---|---|---|---|---|---|---|---|
| PointCLIP | 6.3 | 0.0 | 0.0 | 0.0 | 0.0 | 0.7 | 0.0 | 0.0 | 91.8 | 0.0 | 0.0 | 0.0 | 15.0 | 0.0 | 0.0 | 0.0 | 0.0 | 0.0 |
| PointCLIP V2 | 11.0 | 0.0 | 0.0 | 23.8 | 0.0 | 0.0 | 0.0 | 0.0 | 7.8 | 0.0 | 90.7 | 0.0 | 0.0 | 0.0 | 64.4 | 0.0 | 0.0 | 0.0 |
| CLIP2Point | 24.9 | 20.8 | 0.0 | 85.1 | 43.3 | 26.5 | 69.9 | 0.0 | 20.9 | 1.7 | 31.7 | 27.0 | 0.0 | 1.6 | 46.5 | 0.0 | 22.4 | 25.6 |
| PointCLIP w/ TP. | 26.1 | 0.0 | 55.7 | 72.8 | 5.0 | 5.1 | 1.7 | 0.0 | 77.2 | 0.0 | 0.0 | 51.7 | 0.3 | 0.0 | 0.0 | 40.3 | 85.3 | 49.2 |
| CLIP2Point w/ TP. | 35.2 | 11.8 | 3.0 | 45.1 | 27.6 | 10.5 | 61.5 | 2.6 | 71.9 | 0.3 | 33.6 | 29.9 | 4.7 | 11.5 | 72.2 | 92.4 | 86.1 | 34.0 |
| CLIP[2] | 38.5 | 32.6 | 67.2 | 69.3 | 42.3 | 18.3 | 19.1 | 4.0 | 62.6 | 1.4 | 12.7 | 52.8 | 40.1 | 9.1 | 59.7 | 41.0 | 71.0 | 45.5 |
| **Uni3D** | **45.8** | 58.5 | 3.7 | 78.8 | 83.7 | 54.9 | 31.3 | 39.4 | 70.1 | 35.1 | 1.9 | 27.3 | 94.2 | 13.8 | 38.7 | 10.7 | 88.1 | 47.6 |

under two different training settings. "Ensembled" indicates that the backbones are trained under all the four datasets same as OpenShape and "Ensembled (no LVIS)" further excludes the shapes from the Objaverse-LVIS subset. We justify that even when LVIS shapes are included in the training shapes, i.e., the "Ensembled" dataset, their test-time category labels are probably not included in the training texts. The quantitative comparison is shown in Tab. 1, where Uni3D significantly outperforms the previous state-of-the-art methods under different settings.

## 4.2 FEW-SHOT LINEAR PROBING

Linear probing is a widely used approach for evaluating the learned representation of a model. To evaluate the linear probing ability of Uni3D, we follow the common setting as OpenShape (Liu et al., 2023) to freeze the parameters of Uni3D and only train a linear classifier on few-shot class labels. We conduct few-shot linear probing under the difficult Objaverse-LVIS dataset with labeled training samples per class from 1, 2, 4, 8 to 16. Fig. 4 summarizes the performance of Uni3D in comparison with OpenShape (Liu et al., 2023) (PointBERT backbone and SparseConv backbone), ULIP (Xue et al., 2023a) (official release and the version retrained on the large ensembled dataset) and PointCLIP V2 (Zhu et al., 2022). Uni3D significantly outperforms all the other methods by a large margin under all the few-shot settings.

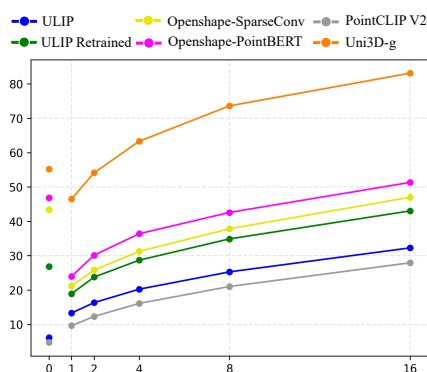

Figure 4: Few-shot linear probing on Objaverse-LVIS. We report the average performance over 10 random seeds.

## 4.3 OPEN-WORLD UNDERSTANDING

To evaluate the capability of Uni3D in 3D understanding of real-world shapes and scenes, we follow CLIP[2] (Zeng et al., 2023) to conduct experiments under ScanNet (Dai et al., 2017) to explore the zero-shot recognition performance of Uni3D under real-world scenarios. Note that the ground truth instant segmentation is available for all the methods and the target is to recognize the category of each instant of the scene in a zero-shot way. ScanNet (Dai et al., 2017) is a popular real-scanned 3D dataset containing 1.5K reconstructed meshes of real-world scenes. We adopt the same setting as CLIP[2] to split classes and evaluate the results under the test set of ScanNet.

We compare our proposed Uni3D with the state-of-the-art methods PointCLIP (Zhang et al., 2022), PointCLIP V2 (Zhu et al., 2022), CLIP2Point (Huang et al., 2022a) and CLIP[2] (Zeng et al., 2023). The quantitative comparison is shown in Tab. 2. "PointCLIP w/TP" and "CLIP2Point w/TP" donate training PointCLIP and CLIP2Point with the real-world data provided by CLIP[2]. Note that "PointCLIP w/TP", "CLIP2Point w/TP" and CLIP[2] are trained under 1.6M triplets of real-world point cloud-image-text samples, while Uni3D is only trained under available synthetic data. Nonetheless, Uni3D achieves the best performance among all the previous methods. The results demonstrate the capability of Uni3D to perform real-world recognition and understanding even without training under real-world data. The reason is that Uni3D distills some perceptions of the real world from the CLIP models which are trained under large-scale real-world images and text. Moreover, by scaling up model size, Uni3D achieves a larger representation bandwidth, leading to superior performance under difficult real-world rscenarios. The qualitative comparison is shown in Fig. 5, where Uni3D produces much more accurate zero-shot recognition results than PointCLIP V2 and CLIP2Point. We do not visually compare with CLIP[2] since its code and model are not publicly available.

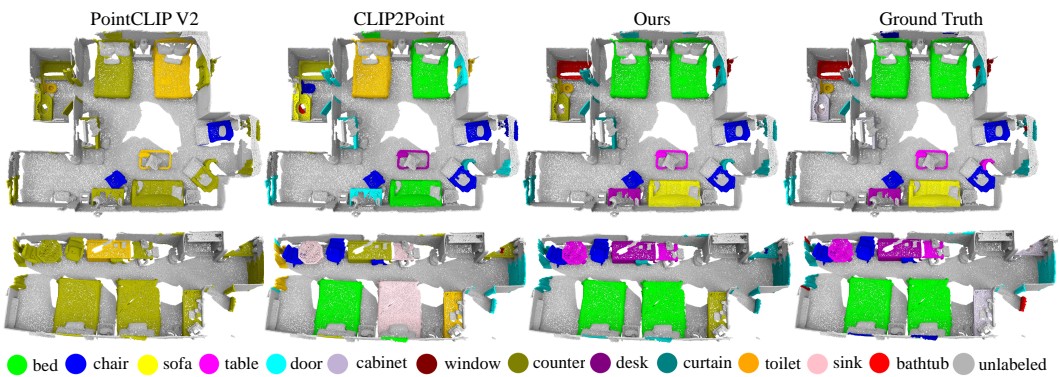

PointCLIP V2    CLIP2Point    Ours    Ground Truth

● bed  ● chair  ● sofa  ● table  ● door  ● cabinet  ● window  ● counter  ● desk  ● curtain  ● toilet  ● sink  ● bathtub  ● unlabeled

Figure 5: Comparisons of real-world zero-shot recognition results on Scannet dataset.

Table 4: Open-vocabulary segmentation results on the ShapeNetPart dataset.

| Method | Seen Categories | | | | | | | | | | | Unseen Categories | | | | | | | |
|---|---|---|---|---|---|---|---|---|---|---|---|---|---|---|---|---|---|---|---|
| | $mIoU_C$ | Car | Knife | Lamp | Moto | Pistol | Rocket | Guitar | Skate | Chair | Cap | Plane | Bag | Earph | Laptop | Mug | Table | $mIoU_C$ | $mIoU_C$-ALL |
| **Uni3D** | **76.0** | 74.1 | 86.3 | 85.7 | 62.7 | 77.8 | 41.7 | 89.3 | 71.0 | 89.9 | 81.8 | 23.4 | 57.1 | 57.1 | 26.2 | 50.0 | 54.4 | **44.7** | **64.3** |

## 4.4 OPEN-VOCABULARY / FEW-SHOT PART SEGMENTATION

Some prior methods (Rao et al., 2022; Yang et al., 2022) have demonstrated that transferring the knowledge gained from image-text contrastive learning, i.e., CLIP, can yield significant performance improvements in 2D dense prediction tasks (e.g. segmentation and detection). However, transferring this knowledge to 3D dense prediction tasks is barely explored. We propose a novel approach for 3D dense prediction with Uni3D, and justify the effectiveness with part segmentation experiment. For more details on the approach, please refer to Sec. B of the appendix.

We conduct part segmentation experiments under ShapeNetPart dataset (Yi et al., 2016). The results in Tab. 3 demonstrate that when supervised with only 1 or 2 samples per class, Uni3D outperforms Point-BERT by +13.3%/+9.8%. Moreover, we largely increase the training samples used for comparative methods to 10% or 20% of the training set. These settings surpass training samples in Uni3D's one-shot or two-shot settings by two orders of magnitude. Even in the face of such a discrepancy in the number

Table 3: Few-shot part segmentation results on the ShapeNetPart dataset.

| Method | Data | $mIoU_C$ | Data | $mIoU_C$ |
|---|---|---|---|---|
| PointNet | | 72.7 | | 73.5 |
| PointNet++ | 10% | 74.8 | 20% | 76.8 |
| PointCNN | train set | 60.4 | train set | 64.1 |
| SSCN | | 60.2 | | 65.2 |
| PointBERT | | 76.4 | | 79.6 |
| PointBERT | 1-shot | 66.2 | 2-shot | 71.9 |
| **Uni3D** | | **75.9** | | **78.2** |

of training samples, Uni3D still achieves comparable performance in terms of overall mIoU. The visual comparisons with PointBERT are provided in Sec. B of the appendix.

"Open-vocabulary part segmentation" quantifies the ability of Uni3D to learn fine-grained semantic information of local point clouds during multi-modal contrastive pre-training. We partition the ShapeNetPart dataset into two subsets: "Seen Categories" and "Unseen Categories." In the "Seen Categories" subset, the text of ground-truth part labels serve as training samples of Uni3D for learning part semantics, while in the "Unseen Categories" subset, the text of ground-truth part labels is unseen during training and is only utilized for testing. The superior performance of Uni3D in Tab. 4 demonstrates its ability to discern fine-grained 3D patterns, even for part-level semantic concepts not encountered in the "Seen Categories". These results robustly affirm Uni3D's capacity to transfer the learned patterns in a close set of 3D parts to open-vocabulary parts, utilizing the rich open-world knowledge distilled from the pre-trained CLIP model. We believe that Uni3D opens avenues to achieve fine-grained, cross-category segmentation of open-vocabulary 3D concepts.

## 4.5 POINT CLOUD PAINTING

We propose to leverage the trained Uni3D for painting point clouds by exploring the learned 3D semantic patterns in Uni3D. Specifically, given an initial point cloud and an input prompt, we optimize the appearance, i.e., RGB channel of the point cloud, by maximizing the cosine similarity between the feature of the point cloud extracted by Uni3D and the feature of the prompt extracted

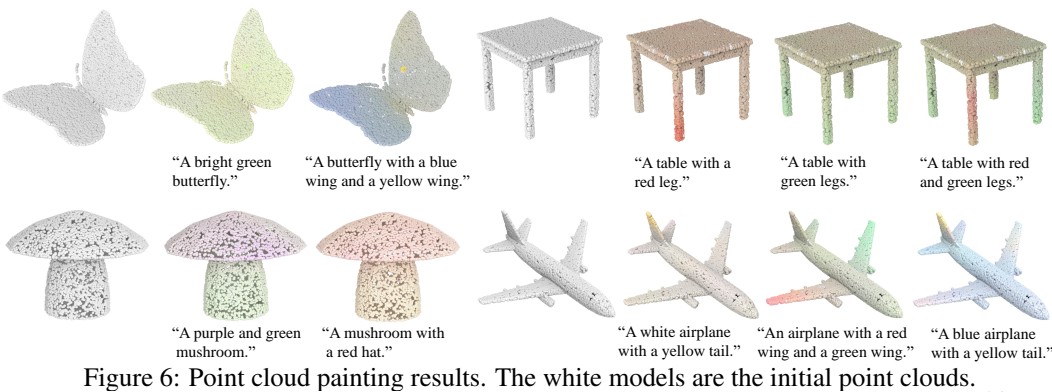

Figure 6: Point cloud painting results. The white models are the initial point clouds.

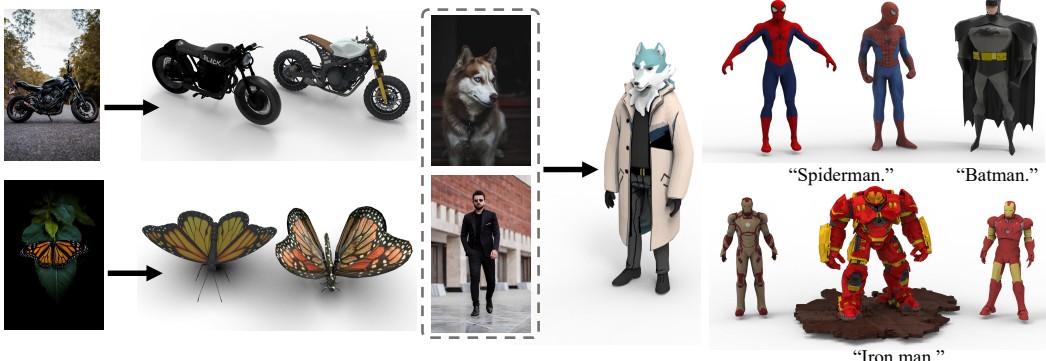

Figure 7: Image-query/text-query 3D shape retrieval results. In the first column, we query two most similar 3D shapes for each query image. In the second column, we take two images as inputs and retrieve the 3D shape similar to both two images. In the third column, we query three most similar 3D shapes for each query text.

with CLIP text encoder. The painting for a point cloud can be achieved within one minute in a single V100 GPU. We show the paintings in Fig. 6, where Uni3D successfully optimizes the point cloud by revealing complex semantics from the prompt. The results demonstrate that Uni3D has learned abundant and diverse 3D patterns via contrastive pretraining.

### 4.6 CROSS-MODAL RETRIEVAL

With the learned multi-modal representations of Uni3D, we can naturally retrieve 3D shapes from images or text. Specifically, we retrieve 3D shapes from the large 3D dataset (Deitke et al., 2023b) by calculating the cosine similarity between the embedding of a query image or a query text prompt and the embedding of 3D shapes. We then perform kNN to achieve the most similar 3D shapes of the query. In Fig. 7, we show that Uni3D successfully retrieves 3D shapes from real-world images. Note that the images for training are only renderings, and there is a big gap between the training images and the real-world images. We also take two images as inputs and retrieve the shape similar to both two images by calculating the cosine similarity between the average of the embedding of two images and the embedding of 3D shapes. The interesting results demonstrate that Uni3D learns a diverse 3D representation with the ability to perceive multiple 2D signals. We further show the results of leveraging Uni3D to retrieve 3D shapes from the input texts in Fig. 7. More visualization results are provided in Sec. C of the appendix.

### 4.7 ABLATION STUDY

We then conduct ablation studies to justify the effectiveness of each design in Uni3D. The default setting is to use the ViT-Base as the backbone with an initialization of EVA (Fang et al., 2022), and the default CLIP teacher is EVA-CLIP-E (Sun et al., 2023). The default data setting is "Ensembled (no-LVIS)". We keep the default experimental setting during ablation studies except for the modified part described in each ablation experiment below.

Table 5: Scaling up model size in Uni3D. $\diamond$ represents the results under "Ensembled" dataset without LVIS shapes. $\dagger$ represents the results under "Ensembled" dataset with LVIS shapes.

| Model | Depth | Width | Heads | #Params | MNet40$^\diamond$ | O-LVIS$^\diamond$ | MNet40$^\dagger$ | O-LVIS$^\dagger$ |
|---|---|---|---|---|---|---|---|---|
| Uni3D-Ti | 12 | 192 | 3 | 6.2M | 85.8 | 43.5 | 85.9 | 46.5 |
| Uni3D-S | 12 | 384 | 6 | 22.6M | 86.0 | 44.8 | 86.0 | 50.6 |
| Uni3D-B | 12 | 768 | 12 | 88.4M | 86.2 | 45.8 | 86.5 | 51.6 |
| Uni3D-L | 24 | 1024 | 16 | 306.7M | 86.6 | 46.2 | 86.6 | 53.2 |
| Uni3D-g | 40 | 1408 | 16 | 1016.5M | 86.8 | 47.2 | 88.2 | 55.3 |

Table 6: Different CLIP teachers at different model scales.

| CLIP variant | Pretrain data | #Params | O-LVIS |
|---|---|---|---|
| EVA-CLIP-B/16 | Merged-2B | 150M | 42.3 |
| OpenAI-CLIP-B/16 | WIT-400M | 150M | 42.7 |
| OpenCLIP-B/16 | LAION-2B | 150M | 43.4 |
| OpenCLIP-bigG/14 | LAION-2B | 2.5B | 44.5 |
| EVA-CLIP-E/14+ | LAION-2B | 5.0B | 45.8 |

Table 7: Initializing Uni3D with different pretrained large models.

| Init variant | O-LVIS |
|---|---|
| None | 44.8 |
| DINO | 45.0 |
| EVA-CLIP | 45.2 |
| EVA | 45.8 |
| EVA + Freeze ViT | 15.7 |

**Scaling Up Model Size.** We first explore the effectiveness of scaling up the model size of Uni3D in Tab. 5. Since we leverage a unified vanilla transformer structurally equivalent to ViT as the foundational 3D representation model, we can simply scale up Uni3D with the well-studied unified 2D/NLP scaling-up strategies. Specifically, we follow the scaling up principles of the plain ViT (Dosovitskiy et al., 2020) to increase parameters from 6 M (Tiny), 23 M (Small), 88 M (Base), 307 M (Large) to 1 B (giant). The hyper-parameters on the model architecture are detailed in Tab. 5. The performance under different model scales demonstrates that scaling up the model size of Uni3D can significantly improve the 3D representation.

**Switching / Scaling Up CLIP Teachers.** We justify that Uni3D is a flexible framework where we can switch the off-the-shelf SoTA CLIP models as the teacher. To this end, we investigate the performances of Uni3D with different CLIP teachers at different scales. Specifically, we evaluate various CLIP models (e.g. OpenAI-CLIP (Radford et al., 2021), OpenCLIP (Cherti et al., 2023) and EVA-CLIP (Sun et al., 2023)), and also explore large scale CLIP models (e.g., OpenCLIP-bigG, EVA-CLIP-E). The quantitative comparison is shown in Tab. 6, with the best performance achieved by the largest CLIP model EVA-CLIP-E. The results show that the capability and model scale of CLIP teachers are key factors for achieving better performance. Moreover, it indicates the potential of Uni3D to progress with the progress of CLIP models by switching state-of-the-art CLIP teachers.

**Initializing Transformer.** We further conduct ablation studies to explore the effectiveness of initializing Uni3D with 2D pretraining or multi-modal large models. In Tab. 7, we report the performance of training Uni3D from scratch (None) and initializing Uni3D with off-the-shelf 2D pretraining model DINO (Caron et al., 2021) / EVA (Fang et al., 2022) and SoTA CLIP model EVA-CLIP (Sun et al., 2023). The best performance is achieved with the SoTA 2D pretraining model EVA (Fang et al., 2022). We also demonstrate that leveraging the frozen parameters from the 2D pretrained ViT model may fail to provide strong 3D understanding without fine-tuning, as shown in "EVA + Freeze ViT" of Tab. 7. For more analysis on initializing Uni3D, please refer to Sec. D of the appendix.

## 5  CONCLUSION

We present Uni3D, a unified framework that scales up a 3D representation model to one billion parameters. We directly leverage a unified transformer structurally equivalent to ViT as the model, which allows us to simply scale up Uni3D with the well-studied unified 2D/NLP scaling-up strategies. Moreover, Uni3D can leverage abundant 2D pretrained models as initialization and image-text aligned models as the target, unlocking the great potential of 2D models and strategies to the 3D world. Uni3D achieves state-of-the-art performance in various 3D understanding tasks including zero-shot and few-shot classification, open-world understanding, part segmentation, etc. We believe that Uni3D can serve as a 3D foundation model to enable many applications in the 3D community.

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

## A    TRAINING DETAILS

We freeze the CLIP text and image encoders while focusing on training the 3D encoder utilizing the cross-modal contrastive loss. We employ the Adam (Kingma & Ba, 2014) optimizer with a peak learning rate of 1e-3 that gradually decreases following a cosine learning rate schedule. To enhance training stability, we adopt stochastic depth (Huang et al., 2016) regularization. We also leverage the FLIP (Li et al., 2023b) technique, which randomly masks 50% of point tokens during training, reducing time complexity by half. We precache text and image CLIP embeddings of all shapes, allowing us to increase the total batch size to 1152 and greatly accelerating training. To further improve the training process, we adopt DeepSpeed (Rasley et al., 2020) with ZeRO stage-1 optimizer and fp16 precision with dynamic loss scaling (Rajbhandari et al., 2020). Taking advantage of the aforementioned strategies, our largest model, i.e., Uni3D-g with one billion parameters, converges in approximately 20 hours with $24 \times$ NVIDIA-A100-SXM4-40GB GPUs.

## B    PART SETMENTATION DETAILS

Some prior methods (Rao et al., 2022; Yang et al., 2022) have demonstrated that transferring the knowledge gained from image-text contrastive learning, i.e., CLIP, can yield significant performance improvements in 2D dense prediction tasks (e.g. segmentation and detection). However, transferring this knowledge to 3D dense prediction tasks is barely explored. We seek to find a way to convert the global point cloud-text alignment learned by Uni3D into a local point-text alignment. We aim to demonstrate that the object-level pre-training in Uni3D is sufficient for learning detailed local 3D visual concepts. Specifically, we select the features from $4^{th}$, $8^{th}$ and the last layer of the ViT in Uni3D, denoted as $H^4$, $H^8$ and $H^{12}$. Following PointNet++ (Qi et al., 2017b), we employ feature propagation to upsample group features $H^4$, $H^8$ and $H^{12}$ into point-wise features. During training, we freeze the Uni3D backbone and only optimize the parameters in the feature propagation layer, with supervision to align point-wise features and text features of ground-truth part labels, which are extracted by the CLIP text encoder. By freezing the parameters of learned Uni3D, we focus on effectively exploring the pre-trained fine-grained knowledge.

The visual comparison in Fig. 8 shows that our method can produce more accurate segmentation results in the one-shot part segmentation setting.

## C    MORE VISUALIZATION OF CROSS-MODAL RETRIEVAL

In Fig. 9, we visualize more 3D shapes retrieved from real-world one or multiple images. We further show the results of leveraging Uni3D to retrieve 3D shapes from the input texts in Fig. 10.

## D    MORE ANALYSIS ON INITIALIZING UNI3D

As demonstrated in Tab. 7, the best performance is achieved by initializing Uni3D with the SoTA 2D pretraining model EVA (Fang et al., 2022). The reason is that EVA model learns powerful

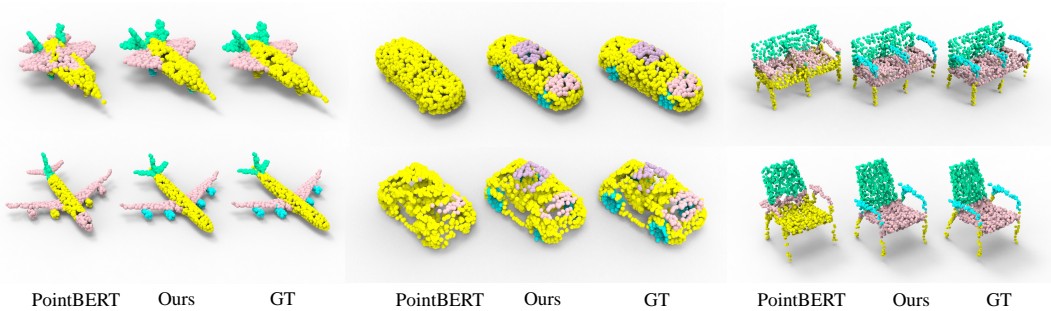

Figure 8: Comparison in one-shot part segmentation under ShapeNetPart.

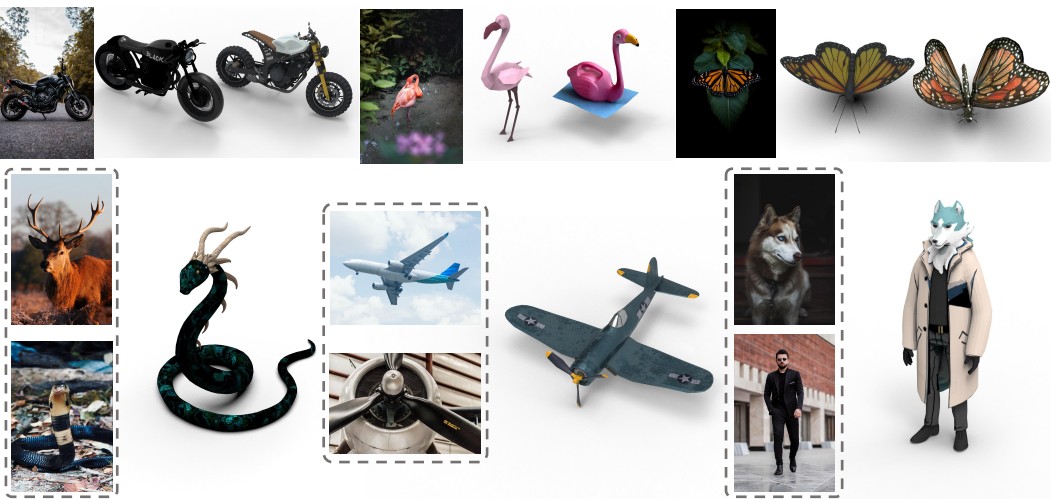

Figure 9: Image-query 3D shape retrieval results. In the first row, we query two most similar 3D shapes for each query image. In the second row, we take two images as inputs and retrieve the 3D shape similar to both two images.

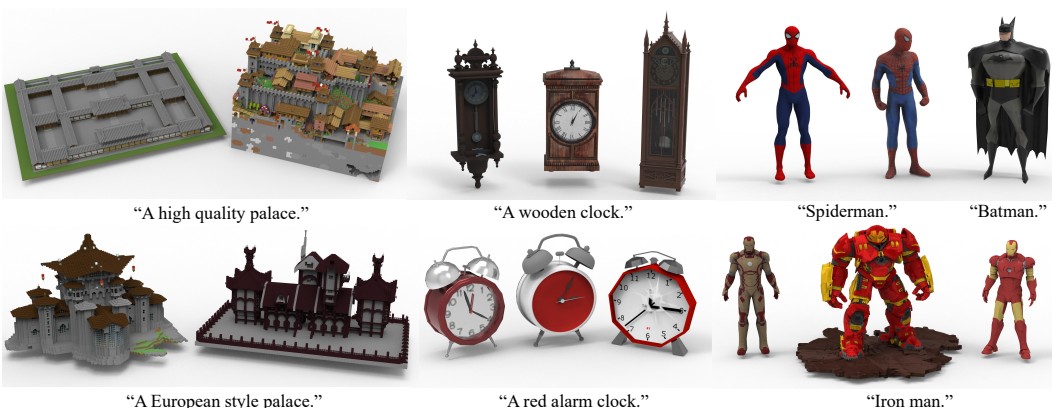

Figure 10: Text-query 3D shape retrieval results.

and general representations to serve as a fine initialization of cross-modal contrastive learning (e.g. CLIP) as demonstrated in EVA (Fang et al., 2022), EVA-02 (Fang et al., 2023) and EVA-CLIP (Sun et al., 2023).

We justify that Uni3D also learns a cross-modal representation similar to CLIP, where the general patterns learned by EVA play a key role in improving and stabilizing the training of Uni3D. The analysis is further supported by the results of two-modal contrastive learning as shown in Tab. 8. Specifically, we conduct experiments to train Uni3D with only contrastive loss with CLIP image features (+only image) or CLIP text features (+only text), respectively. The results show that the optimization crashed without EVA initialization in the difficult situation where only images are available (20.7 vs. 40.1) or only texts are available (12.4 vs. 26.3).

Table 8: Effect of initializing Uni3D under two-modal situation.

| Init variant | O-LVIS |
|---|---|
| None Init + only text | 20.7 |
| None Init + only image | 12.4 |
| EVA + only text | 40.1 |
| EVA + only image | 26.3 |

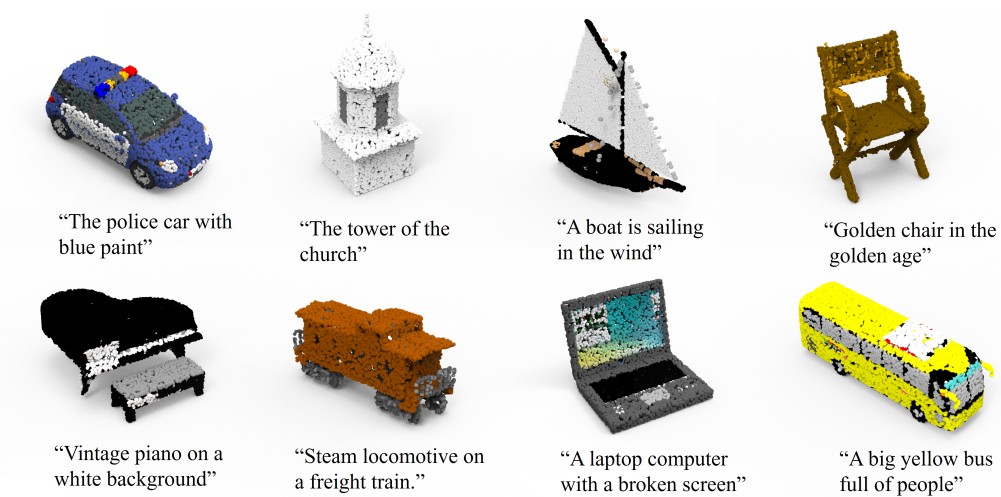

"The police car with blue paint"   "The tower of the church"   "A boat is sailing in the wind"   "Golden chair in the golden age"

"Vintage piano on a white background"   "Steam locomotive on a freight train."   "A laptop computer with a broken screen"   "A big yellow bus full of people"

Figure 11: Point cloud captioning results. We show the input point clouds with the generated captions under them.

Table 9: Few-shot linear probing on Objaverse-LVIS. We report the average Top1 accuracy over 10 random seeds.

| Objaverse-LVIS | 1-shot | 2-shot | 4-shot | 8-shot | 16-shot |
|---|---|---|---|---|---|
| PointCLIP V2 | 9.6 | 12.3 | 16.1 | 21.0 | 27.9 |
| ULIP | 13.3 | 16.3 | 20.3 | 25.3 | 32.3 |
| ULIP Retrained | 18.9 | 23.8 | 28.7 | 34.9 | 43.0 |
| OpenShape-PointBERT | 24.0 | 30.1 | 36.4 | 42.6 | 51.3 |
| OpenShape-SparseConv | 21.2 | 25.8 | 31.2 | 37.8 | 47.0 |
| **Uni3D** | **46.5** | **54.1** | **63.3** | **73.6** | **83.1** |

# E  FEW-SHOT RESULTS

We conduct few-shot linear probing under the difficult Objaverse-LVIS dataset with labeled training samples per class from 1, 2, 4, 8 to 16. The comparison is shown in Fig. 4. We further provide the detailed quantitative results in Tab. 9.

# F  POINT CLOUD CAPTIONING WITH UNI3D

With the learned 3D representations which are aligned with text/image embeddings, we can further apply Uni3D to the point cloud captioning task with CLIP-based image captioning approaches. Specifically, we follow OpenShape (Liu et al., 2023) to feed our 3D shape embeddings to CLIPCap (Mokady et al., 2021) for performing point cloud captioning. We directly employ the off-the-shelf CLIPCap model without any fine-tuning. CLIPCap adopts the image embeddings achieved with OpenAI-CLIP-B/32 for captioning, therefore, we train a variant of Uni3D with the OpenAI-CLIP-B/32 teacher to align with CLIPCap. The visualizations of captioning 3D point clouds with Uni3D are shown in Fig. 11.

