# OpenReview forum: "Uni3D: Exploring Unified 3D Representation at Scale"
_ICLR.cc/2024/Conference — ICLR 2024 spotlight_

### Official Review · Reviewer_1Xtq · 2023-10-31

**Soundness:** 3 good
**Presentation:** 3 good
**Contribution:** 2 fair
**Rating:** 6
**Confidence:** 4

**Summary:**

The paper presents Uni3D, a general 3D foundation model to explore the "unified" 3D representation at scale. Given the point cloud at input, it uses a 2D initialized ViT to align 3D / 2D features. It scales up to 1B (which is very large for 3D tasks) and achieves good performance on a broad range of tasks.

**Strengths:**

+ Simple and effective structure to scale up the model capacity to 1B. This is very impressive in the 3D point cloud domain.

+ Good performance in a wide span of tasks with detailed experimental results.

**Weaknesses:**

- The exact insight as to why the proposed method succeeds in 3D domain is not fully stated or analyzed. Please see the questions below.

**Questions:**

1. The successful scaling up to 1B parameters is a very key contribution from this work. It works well on multiple downstream tasks. Does the gain come from the 2D pretrained models? Table 7 seems to give some ablations and yet this is not clearly stated in the introduction. I was wondering this since it would tell people whether to focus more on 2D data/pretraining to resolve 3D problems, or 3D pretraining is essential. If I am getting this right, using none 2D initialized weights as shown in Table 7, the gain is not too much obvious (44.8 vs 45.8).

2. Uni3D is verified on a wide variety of tasks and benchmarks. This is motivating. Do you plan to try some challenging and realistic settings, eg. autonomous driving settings with point clouds? That would strength the proposed approach to great extent.

3. The last row in Table 1 shows the result of models separately trained on each benchmark. The unified approach of Uni3D is on par with them, demonstrating the generalization or universality. Does Uni3D potentially could surpass the performance of the model trained each on one particular benchmark? I was wondering the nessecity of training a universal model.

---
Minor:
- Typo in Figure 1, "ensambled"

---

> ### Author Response · Authors · 2023-11-20
> **Responses to Reviewer 1Xtq (1/2)**
>
> We sincerely appreciate Reviewer 1Xtq for the acknowledgment of our work and constructive feedback. We will respond comprehensively to the questions as follows.
>
> **Q1: Where does the gain of scaling up Uni3D to billion-scale parameters come from.**
>
> The performance gain of Uni3D in scaling up model sizes mainly comes from two parts:
>
> **1. The well-explored scaling up strategies from the unified 2D ViT model.** We directly leverage the vanilla transformer structurally equivalent to ViT, which can naturally solve the difficulties in scaling designs in 3D domain by simply scaling up the model size with the well-studied unified 2D/NLP scaling-up strategies. The effectiveness and efficiency of our scaling-up strategy are fully demonstrated by the comprehensive exploration of scaling up ViT in the 2D vision domain. As shown in Fig. 1 and Tab. 5, we observe continuous performance improvements as the scaling of model sizes under the flexible and unified framework.
>
> **2. The sharable 2D prior from large pre-trained 2D ViT.**
> Uni3D directly leverages the vanilla transformer structurally equivalent to ViT as the 3D backbone, which brings a new perspective of introducing pretrained priors. Specifically, we can naturally adopt the pretrained large models in other modalities which share the same vanilla transformer as ours to initialize Uni3D. These pretrained models are trained in datasets consisting of billions of images and texts, which already learn rich underlying representational abilities for Transformer and have the potential to enhance and stabilize the learning of large-scale 3D representations.
>
> To demonstrate the key role of the large pre-trained 2D ViT for initializing Uni3D, we conduct extensive ablation studies under different scales of Uni3D from 6 M (Tiny), 23 M (Small), 88 M (Base), 307 M (Large) to 1 B (giant). We keep the default data setting the same as the ablation studies of the main paper, i.e., “Ensembled (no-LVIS)”. We list the results below:
>
>  ***Table A : Ablation studies on the effect of initializing Uni3D with pre-trained 2D ViT.***
> | Model Size | From scratch | 2d pretrained ViT init |
> | --- | --- | --- |
> | Tiny | 42.8 | 43.5 |
> | Small | 43.7 | 44.8 |
> | Base | 44.8 | 45.8 |
> | Large | 45.0 | 46.2 |
> | giant | 26.6（broken） | 47.2 |
>
> As shown, the initialization from large pre-trained 2D ViT improves the performance at all model sizes. Specifically, the training of the model with extremely large scales of parameters (e.g. 1 B of giant) is **broken** when optimizating from scratch, while the continuous improvement is further achieved by introducing 2D priors from pre-trained large vision models. This justifies that the powerful initialization from 2D pretrained models is necessary for achieving performance gain in scaling up Uni3D model to extremly large parameters (e.g. 1B).
>
> We further refer the review to Tab. 10 of the main paper where we conducted ablation studies under two-modal alignment. The results show that the optimization crashed without EVA initialization in the difficult situation where only images are available (40.1->20.7) or only texts are available (26.3->12.4). The results in Tab. 10 of the main paper are reported under the Uni3D-Base model. We additionally supplement this ablation with different model scales, i.e. from 6 M (Tiny), 23 M (Small) to 88 M (Base), under the difficult text-point cloud alignment situation. We report the results below:
>
>   ***Table B : Learning text-point cloud alignment from scratch or initializing with pre-trained 2D ViT.***
> | Model Size | From scratch | 2d pretrained ViT init |
> | --- | --- | --- |
> | Tiny | 37.0 | 38.3 |
> | Small | 37.8 | 39.6 |
> | Base | 20.7（broken） | 40.1 |
>
>
> We observe the same experimental phenomena as Table A, where the large pre-trained 2D ViT improves the performance at all model sizes. More specifically, the performance of models trained from scratch improves normally from Tiny to Small (37.0 -> 37.8), but crashes when further scaling up from Small to Base (37.8->20.7). While models trained with 2D priors of pre-trained large vision models obtain continuous improvements from Tiny, Small, to Base (38.3->39.6->40.1).
>
> These two ablation studies demonstrate the necessity of introducing 2D priors from large pre-trained 2D ViT for initializing Uni3D. The 2D priors play the key role in improving and stablizing the optimization of Uni3D models, especially when scaling up to large parameters, where the optimization crashed when training from scratch. Moreover, we observe that for the more difficult situation, such as text-point cloud alignment, the crashment comes eailer during the model size scaling. And the 2D prior stabilizes all the situations and achieves continuous improvements in all the settings.

---

> ### Author Response · Authors · 2023-11-20
> **Responses to Reviewer 1Xtq (2/2)**
>
> **Q2: Expending Uni3D to more challenging settings, e.g., autonomous driving.**
>
> We believe that Uni3D, which has powerful and universial 3D perceptions, has great potentials in some challenging and realistic settings, eg. autonomous driving settings with point clouds, VR/AR and robotics. We consider these as the exciting furture work for Uni3D and will be committed to exploring the application prospects of Uni3D.
>
> **Q3: The performance of training a universal Uni3D model.**
>
> We justify that all the experiment results on Tab. 1 are tested in a zero-shot way with the universial Uni3D model trained under two data settings, i.e., Ensembled (no LVIS) and Emsembled. The last row reports the best results achieved on different benchmarks under different convergences of optimization. Specifically, the best performance of ModelNet and ScanObjNN is achieved with the early stopped Uni3D model since those two benchmarks are quite easier with few categories and samples, where more training iterations will lead to performance degeneration. While the best performance of Objaverse-LVIS is achieved with the fully converged Uni3D model since Objaverse-LVIS is much more difficult with 1,156 categories containing some out-of-the-distribution samples, which requires more extensive training.

---

> ### Author Response · Authors · 2023-11-29
> **Looking Forward to Your Participation as Reviewer-Author Discussion Period has been Extended**
>
> We express our sincere gratitude to the reviewer for the invaluable feedback and time invested in evaluating our work. The Reviewer-Author discussion period for our submission Uni3D has been extended to December 1st due to the extra emergency review. Therefore, we would like to inquire if our responses have adequately addressed your questions. And we anticipate your participation in the extended Reviewer-Author discussion phase, as your insights are invaluable for refining and enhancing the quality of our paper. If you have any additional queries or require further clarification, please do not hesitate to inform us. We are fully committed to addressing any concerns to ensure a comprehensive resolution.  Thank you sincerely for your time and consideration.
>
> As a supplement to the **“Q1: Where does the gain of scaling up Uni3D to billion-scale parameters come from.”** We further provide extra experiments and analysis on the key design to leverage sharable 2D prior from large pre-trained 2D ViT by conducting experiments under a much smaller dataset containing 20% data samples from the "Ensembled" dataset. We list the results with Uni3D-Base as the backbone below:
>
> ***Table D: The impact of ViT initialization under a small dataset (20% data samples).***
> | Uni3D-Base | From scratch | 2d pretrained ViT init |
> | --- | --- | --- |
> | Objaverse-LVIS | 38.6 | 42.5 |
> | ModelNet40 | 80.6 | 83.5 |
> | ScanObjectNN | 52.6 | 60.2 |
>
>
>
> As shown, initializing Uni3D with pretrained 2D ViT demonstrates more significant improvements on the much smaller data scale. Specifically, the initialization brings an improvement of about 4% in Objaverse-LVIS, 3% in ModelNet40 and 7.6% in ScanObjectNN. The results demonstrate that the 2D pretrained prior plays a more important role with the limited data scale and further highlight the effectiveness of the unified ViT architecture of Uni3D, which receives sharable 2D priors from large pre-trained 2D ViT.

---

### Official Review · Reviewer_8Neg · 2023-11-04

**Soundness:** 3 good
**Presentation:** 3 good
**Contribution:** 3 good
**Rating:** 8
**Confidence:** 4

**Summary:**

This paper introduces a 3D foundation model, dubbed Uni3D, which uses a 3D point tokenizer and ViT to align 3D features with CLIP features (images and texts). Uni3D is trained with the triplet contrastive loss OpenShape used. By scaling up the model size of ViT as a point encoder to a billion-scale, Uni3D shows impressive performance on zero-shot and few-shot 3D perception tasks including classification and semantic segmentation. Although the experiment results are impressive, the technical novelty of the proposed is limited, and a few analyses (as described in the weakness section) seem necessary to strengthen the paper.

**Strengths:**

[Method] Training a 3D foundation model is a timely topic and the proposed method Uni3D showed impressive results on various benchmarks.

[Experiments] The paper provides not only common benchmarks for open-world understanding but also interesting analysis (e.g., point cloud painting). Especially, the point cloud coloring analysis shows that the trained 3D encoder can encode the color of a 3D point cloud, which is aligned with text features.

[Detailed explanation] The paper provides implementation details to help readers understand the proposed method well. For example, I could understand that Uni3D used a PointNet++-based upsampling strategy from the details related to part segmentation experiments, as shown in the Appendix.

**Weaknesses:**

[Novelty] The proposed method mainly follows the previous work, OpenShape, in terms of the training data construction (Sec 3.2) and the training objective (Eq. (1)). From my understanding, the only difference is that Uni3D uses PointBERT’s 3D tokenizer + ViT as a 3D encoder while OpenShape uses PointBERT. Although OpenShape is still in arXiv, I recommend the authors clarify what differentiates Uni3D from OpenShape since OpenShape is the most relevant baseline and high similarity to this work.

[Experiments] Although the paper provides text-to-3D retrieval and image-to-3D retrieval, I think 3D-to-text (captioning) and 3D-to-image (generation) experiments need to be included in the paper to show the good alignment of 3D, image, and text.

[Analysis] As shown in Figure 1, scaling up OpenShape does not improve its zero-shot accuracy, unlike Uni3D. Why can Uni3D have such consistent improvement while it has a similar architecture to OpenShape? Does this improvement come from the initialization of the large pre-trained 2D ViT? I recommend the authors provide an analysis of this to make the paper stronger.

**Questions:**

Please refer to the weakness section.

---

> ### Author Response · Authors · 2023-11-20
> **Responses to Reviewer 8Neg (1/2)**
>
> We deeply appreciate the reviewer 8Neg for the thoughtful feedback and time invested in evaluating our work. We respond to each question below.
>
> **Q1: The contributions of Uni3D and differences to OpenShape.**
>
> We justify that the training objective to align text-image-point cloud representations is not considered as a technical contribution of Uni3D, nor is it for OpenShape. The objective was first introduced in ULIP [1], and both OpenShape and Uni3D follow the training objective and focus on scaling up 3D representations. While OpenShape focuses on the data scaling to leverage the large open-source Objaverse dataset with specific efforts on ensembling multiple datasets and data augmentation to filter and enrich text descriptions. We go beyond OpenShape and propose to focus on the scaling up on model sizes. We achieve this by designing unified 3D representation to directly adopt the vanilla transformer structurally equivalent to ViT as the 3D backbone, which leverages abundant 2D pretrained models as initialization. This unlocks the great potential of 2D models and scaling-up strategies to the 3D world, which plays the key role in scaling up Uni3D to billion-scale parameters.
>
> We summarize the main difference between Uni3D and OpenShape as below:
>
> 1. We design the first billion-scale 3D foundation model, and demonstrate a billion-scale 3D representation model can transfer well to various downstream tasks and scenarios. This is acheived by introducing the sharable 2D prior and scaling up strategies from large pre-trained 2D ViT as we will discuss in **Q3**. On the contrary, OpenShape struggles to scale up models to large parameters, which shows degradation during the model size scaling up (e.g. 72 M in Fig. 2).
> 2. We unify the 3D representation backbone to a 2D ViT. This enables Uni3D to explore the great potential of pretrained priors and scaling-up strategies of large vision models to the 3D world. On the contrary, OpenShape adopts the specifically designed point cloud backbones, e.g., SparseConv, PointBERT, PointMLP, etc. OpenShape achieved inconsistent performance with different backbones. For instance, selecting PointBERT as the 3D backbone performs the best on Objaverse-LVIS, while leveraging SparseConv as the backbone significantly outperforms PointBERT under ScanObjectNN. Uni3D achieves significant improvement under all benchmarks consistently with the unified model.
> 3. We explore more representative 3D tasks and introduce more applications with the aligned point cloud-text-image representations from Uni3D, such as point cloud painting, open-world understanding and the difficult dense prediction task of part segmentation.
>
>
> **Q2: More experiments on the applications of Uni3D.**
>
> We provide the extra application of Uni3D for 3D captioning in Sec.F in the Appendix of the revised paper. More applications (e.g. point cloud-based image generation) will be further included in the next revision.
>
> [1] Xue L, Gao M, Xing C, et al. ULIP: Learning a unified representation of language, images, and point clouds for 3D understanding. CVPR 2023

---

> ### Author Response · Authors · 2023-11-20
> **Responses to Reviewer 8Neg (2/2)**
>
> **Q3: How does Uni3D successfully scale up to billion-scale parameters.**
>
> The success of Uni3D in scaling up model sizes mainly comes from two parts:
>
> **1. The well-explored scaling up strategies from the unified 2D ViT model.** We directly leverage the vanilla transformer structurally equivalent to ViT, which can naturally solve the difficulties in scaling designs in 3D domain by simply scaling up the model size with the well-studied unified 2D/NLP scaling-up strategies. The effectiveness and efficiency of our scaling-up strategy are fully demonstrated by the comprehensive exploration of scaling up ViT in the 2D vision domain. As shown in Fig. 1 and Tab. 5, we observe continuous performance improvements as the scaling of model sizes under the flexible and unified framework.
>
> **2. The sharable 2D prior from large pre-trained 2D ViT.**
> Uni3D directly leverages the vanilla transformer structurally equivalent to ViT as the 3D backbone, which brings a new perspective of introducing pretrained priors. Specifically, we can naturally adopt the pretrained large models in other modalities which share the same vanilla transformer as ours to initialize Uni3D. These pretrained models are trained in datasets consisting of billions of images and texts, which already learn rich underlying representational abilities for Transformer and have the potential to enhance and stabilize the learning of large-scale 3D representations.
>
> To demonstrate the key role of the large pre-trained 2D ViT for initializing Uni3D, we conduct extensive ablation studies under different scales of Uni3D from 6 M (Tiny), 23 M (Small), 88 M (Base), 307 M (Large) to 1 B (giant). We keep the default data setting the same as the ablation studies of the main paper, i.e., “Ensembled (no-LVIS)”. We list the results below:
>
>  ***Table A : Ablation studies on the effect of initializing Uni3D with pre-trained 2D ViT.***
> | Model Size | From scratch | 2d pretrained ViT init |
> | --- | --- | --- |
> | Tiny | 42.8 | 43.5 |
> | Small | 43.7 | 44.8 |
> | Base | 44.8 | 45.8 |
> | Large | 45.0 | 46.2 |
> | giant | 26.6（broken） | 47.2 |
>
> As shown, the initialization from large pre-trained 2D ViT improves the performance at all model sizes. Specifically, the training of the model with extremely large scales of parameters (e.g. 1 B of giant) is **broken** when optimizating from scratch, while the continuous improvement is further achieved by introducing 2D priors from pre-trained large vision models. This justifies that the powerful initialization from 2D pretrained models is necessary for achieving performance gain in scaling up Uni3D model to extremly large parameters (e.g. 1B).
>
> We further refer the review to Tab. 10 of the main paper where we conducted ablation studies under two-modal alignment. The results show that the optimization crashed without EVA initialization in the difficult situation where only images are available (40.1->20.7) or only texts are available (26.3->12.4). The results in Tab. 10 of the main paper are reported under the Uni3D-Base model. We additionally supplement this ablation with different model scales, i.e. from 6 M (Tiny), 23 M (Small) to 88 M (Base), under the difficult text-point cloud alignment situation. We report the results below:
>
>   ***Table B : Learning text-point cloud alignment from scratch or initializing with pre-trained 2D ViT.***
> | Model Size | From scratch | 2d pretrained ViT init |
> | --- | --- | --- |
> | Tiny | 37.0 | 38.3 |
> | Small | 37.8 | 39.6 |
> | Base | 20.7（broken） | 40.1 |
>
>
> We observe the same experimental phenomena as Table A, where the large pre-trained 2D ViT improves the performance at all model sizes. More specifically, the performance of models trained from scratch improves normally from Tiny to Small (37.0 -> 37.8), but crashes when further scaling up from Small to Base (37.8->20.7). While models trained with 2D priors of pre-trained large vision models obtain continuous improvements from Tiny, Small, to Base (38.3->39.6->40.1).
>
> These two ablation studies demonstrate the necessity of introducing 2D priors from large pre-trained 2D ViT for initializing Uni3D. The 2D priors play the key role in improving and stablizing the optimization of Uni3D models, especially when scaling up to large parameters, where the optimization crashed when training from scratch. Moreover, we observe that for the more difficult situation, such as text-point cloud alignment, the crashment comes eailer during the model size scaling. And the 2D prior stabilizes all the situations and achieves continuous improvements in all the settings.

---

> > ### Comment · Reviewer_8Neg · 2023-11-21
> > **Thank you for the responses.**
> >
> > I appreciate the detailed answers as well as the additional experiments. I am more convinced about the value of the paper. Therefore, I calibrate my rating to "accept".

---

> > > ### Author Response · Authors · 2023-11-22
> > > **Thanks to Reviewer 8Neg**
> > >
> > > Dear Reviewer 8Neg,
> > >
> > > Many thanks for all the helpful comments and positive assessment. We really appreciate your expertise and the score upgrade.
> > >
> > > Best,
> > >
> > > Authors

---

### Official Review · Reviewer_7VZo · 2023-11-22

**Soundness:** 3 good
**Presentation:** 3 good
**Contribution:** 3 good
**Rating:** 8
**Confidence:** 5

**Summary:**

**Note: this is an emergency review**

This paper proposes Uni3D, a 3D point cloud foundation model for open-world understanding that achieves state-of-the-art performance on zero-shot shape classification. The Uni3D architecture, drawing inspirations from the scalability of ViT, is composed of a network that encodes local point cloud patches into features, akin to the image patch features in ViT. Subsequently, the transformer layers of ViT process these patch features and output a final global feature. These transformers can be loaded from pretrained 2D visual encoders such as the ones in DINO or EVA-CLIP. Besides demonstrating the superior open-world performance of Uni3D, the authors also showcase applications such as point cloud painting and cross-modal retrieval.

**Strengths:**

- The paper is generally well-written.
- Authors plan to release all the code and pretrained models, which will greatly facilitate future research efforts.
- The proposed ViT-like architecture is intuitive and well-motivated. The architecture also achieves quite a significant performance gain over existing models in e.g., OpenShape.

**Weaknesses:**

- Further exploration and analysis of Uni3D's behaviors could provide additional insights and understanding. Please refer to the "Questions" section below for more details.
- (New concern on Nov 22) Prior works like OpenShape have been trained on a much smaller batch size than Uni3D (OpenShape is trained using a batch size of 200 on a single A100-80G, while Uni3D is trained using a batch size of 1152 on 24x A100-40G). My question is, if authors train Uni3D using the same batch size as OpenShape, or train OpenShape using the same batch size as Uni3D, does Uni3D still outperform OpenShape (when both models have similar numbers of parameters)? This experiment will reveal whether the better training setting or the proposed architecture plays a bigger role in the superior performance of Uni3D. Though, regardless of the final findings from this experiment, it won't affect my positive view of the paper.

**Questions:**

- It would be helpful to include a comparison of the inference speed of Uni3D compared to prior work on open-world 3D understanding.
- The main source of performance gains of Uni3D over existing work seems to come from the proposed architecture, instead of how the architecture is initialized (according to authors' reply to Reviewer 1Xtq, whether to initialize the architecture from pretrained ViT only leads to about 1% performance difference). This observation might be attributed to the extensive size of the pretraining dataset, which encompasses 1 million 3D shapes. This raises a question: If a smaller pretraining dataset were used, would initializing with a pretrained ViT have a more significant impact on open-world generalization?

---

> ### Comment · Reviewer_7VZo · 2023-11-22
>
> (Nov 22 update) Hi authors, I have added an additional question in the updated review. Thanks!

---

> ### Author Response · Authors · 2023-11-25
> **Responses to Reviewer 7VZo (1/2)**
>
> We sincerely appreciate the reviewer 7VZo for the invaluable feedback and time invested in evaluating our work. We respond to each question below.
>
> **Q1: Inference speed of Uni3D compared to prior work.**
>
> We provide a comprehensive comparison on the inference speed of Uni3D and prior works. The times are listed below:
>
> ***Table C: Inference speed comparison between Uni3D and prior works.***
> | Model  | Inference time (ms) |
> | --- | --- |
> | Uni3D-Tiny | 6.3 |
> | Uni3D-Small | 7.6 |
> | Uni3D-Base | 10.2 |
> | Uni3D-Large | 16.0 |
> | Uni3D-giant | 20.3 |
> | Openshape-SparseConv | 17.8 |
> | Openshape-PointBERT | 25.2 |
> | ULIP-PointBERT | 58.5 |
>
> The inference time here indicates the time to achieve an embedding for a single point cloud (batch size = 1). All the results are tested under one single V100 GPU. The results demonstrate that Uni3D, which performs the best with the largest model sizes, achieves faster inference times compared to previous works like OpenShape-PointBERT and ULIP-PointBERT. The ViT-based Uni3D also achieves a comparable inference speed to OpenShape-SparseConv, where the SparseConv is a specially designed framework for efficient point cloud processing.
>
> The superior efficiency of Uni3D can be attributed to the efforts in the implementation improvement of Uni3D, especially in the point tokenizer. Specifically, ULIP-PointBERT implements the farthest point sampling (FPS) in point  tokenizer in pure pytorch, while OpenShape-PointBERT improves the implementation of FPS with DGL (Deep Graph Library) and achieves a faster inference time than the pure pytorch implementation of ULIP-PointBERT. We take a further step to leverage a CUDA-accelerated implementation of farthest point sampling provided by PointNet++ [1], which is highly efficient with underlying C++ operations, to implement our point  tokenizer. The efforts in improving point  tokenizer implementation lead to fast inference speed of Uni3D, where even the Uni3D-giant with 1B (1,016M) parameters achieves a speed on par with OpenShape-PointBERT with 32M parameters and OpenShape-SparseConv with 41M parameters.
>
> [1] Qi C R, Yi L, Su H, et al. Pointnet++: Deep hierarchical feature learning on point sets in a metric space. NeurIPS 2017.
>
> **Q2: The impact of initialization with a pretrained ViT under a smaller dataset.**
>
> We greatly thank reviewer 7VZo for the insightful advice on exploring the impact of initializing Uni3D with pretrained ViTs. We follow the suggestions to conduct experiments under a much smaller dataset containing 20% data samples from the "Ensembled" dataset. We list the results with Uni3D-Base as the backbone below:
>
> ***Table D: The impact of ViT initialization under a small dataset (20% data samples).***
> | Uni3D-Base  | From scratch | 2d pretrained ViT init |
> | --- | --- | --- |
> | Objaverse-LVIS | 38.6 | 42.5 |
> | ModelNet40 | 80.6 | 83.5 |
> | ScanObjectNN | 52.6 | 60.2 |
>
>
> As shown, initializing Uni3D with pretrained 2D ViT demonstrates more significant improvements on the much smaller data scale. Specifically, the initialization brings an improvement of about 4% in Objaverse-LVIS, 3% in ModelNet40 and 7.6% in ScanObjectNN. The results demonstrate that the 2D pretrained prior plays a more important role with the limited data scale and further highlight the effectiveness of the unified ViT architecture of Uni3D, which receives sharable 2D priors from large pre-trained 2D ViT.
>
> We furthest justify that the initialization from large pre-trained 2D ViT contributes more than enhancing performance. As shown in *Table A* and *Table B* of the "*Responses to Reviewer 1Xtq (1/2)*", the 2D prior from large pre-trained ViT also stablizes the training of Uni3D, especially when scaling up to extremely large parameters. Without the initialization, the training is **crashed** under the large model size of Uni3D-giant containing 1B parameters. The results demonstrate that the 2D prior from large pre-trained ViT plays a key role in the success of Uni3D to scale up 3D representations to billion-scale parameters.

---

> ### Author Response · Authors · 2023-11-25
> **Responses to Reviewer 7VZo (2/2)**
>
> **Q3: Train Uni3D using the same batch size as OpenShape under similar model parameters.**
>
> We conduct ablation studies and report the performance of training Uni3D-Small (22.6M parameters) with the same batch size as OpenShape-PointBERT (i.e. 200) below:
>
> ***Table E: Ablation studies on the batch sizes.***
>
> |Uni3D-Small | Batch size 200 | Batch size 1152|
> | --- | --- | --- |
> |Objaverse-LVIS| 42.6 | 44.8 |
>
> The experiments are conducted on a single A100-40G GPU under the “Ensembled (no-LVIS)” dataset. The results show that a larger batch size leads to a better performance with about 2% improvement on accuracy. However, even with the much smaller batch size same as OpenShape, Uni3D-Small still achieves significantly better performance than OpenShape-PointBERT (42.6 vs. 39.1) and OpenShape-SparseConv (42.6 vs. 37.0). Also notice that we choose the Uni3D-Small as the backbone here, which only contains 22.6M parameters less than OpenShape-PointBERT (32.3M) and OpenShape-SparseConv (41.3M). Even so, we still outperform OpenShape with the proposed Uni3D backbone unifying to ViT and the robust initialization from large pre-trained 2D ViT.

---

> > ### Comment · Reviewer_7VZo · 2023-11-25
> >
> > Thanks authors for the rebuttal! The additional experiments have enhanced the paper, and I've increased my review confidence.

---

> > > ### Author Response · Authors · 2023-11-26
> > > **Thanks to reviewer 7VZo**
> > >
> > > Dear Reviewer 7VZo,
> > >
> > > Many thanks for all the invaluable feedback and positive assessment. We really appreciate your expertise and the confidence upgrade.
> > >
> > > Best,
> > >
> > > Authors

---

### Author Response · Authors · 2023-11-20
**Response to all: Thank you very much for the thorough reviews.**

We are grateful to the reviewers for their invaluable feedback and the time they dedicated to evaluating our work. We are delighted that reviewers appreciated the representation and the significance of the paper. We are encouraged with the recognition of both reviewers that our work "showed impressive results on various benchmarks" and "is very impressive in the 3D point cloud domain".

We respond to each reviewer separately with detailed analysis, ablation studies and visualizations to solve all the raised questions. Thank you again for your insightful feedback and we are looking forward to continuing the discussion.

---

### Meta-Review · Area_Chair_t7c8 · 2023-12-11

**Metareview:**

The paper presents an effort to scale up the representation of 3D. Results show the effectiveness of the work by outperforming state-of-the-art on various tasks. All reviewers are positive about the contributions and two are highly enthusiastic. The rebuttal is well-accepted by reviewers. The meta-reviewer recommends accepting the submission.

**Justification For Why Not Higher Score:**

The choice of techniques is generally straight-forward.

**Justification For Why Not Lower Score:**

The efforts are significant, and the outcome is very strong. It marks a new progress of 3D representation learning.

---

### Decision · Program_Chairs · 2024-01-16

Accept (spotlight)